# A Preliminary Study on the Effects of Taurine-Enriched Rotifers on the Growth and Survival of the Small Yellow Croaker *Larimichthys polyactis* Larvae

**DOI:** 10.3390/ani12111403

**Published:** 2022-05-30

**Authors:** Jeong-Hyeon Cho, Jae-Hoon Kim, Jin Woo Park

**Affiliations:** Jeju Fisheries Research Institute, National Institute of Fisheries Science, Jeju 63068, Korea; cjh0123@korea.kr (J.-H.C.); hun513@naver.com (J.-H.K.)

**Keywords:** yellow croaker, rotifer, taurine enrichment, larval growth, survival

## Abstract

**Simple Summary:**

Rotifers may not meet the nutritional requirements of cultured marine fish species during seed production. Therefore, we evaluated the effect of feeding rotifers enriched with taurine on the growth performance and survival of the small yellow croaker *Larimichtys polyactis* larvae. The taurine content in the fish increased with the increase in the rotifer taurine levels. The growth and survival rates of the larvae fed taurine-enriched rotifers were higher than those fed control rotifers.

**Abstract:**

The effect of feeding with taurine-enriched rotifers on larval growth and survival in the small yellow croaker *Larimichthys polyactis* was investigated. Rotifers, control (without taurine enrichment) or enriched with a commercial taurine supplement at two concentrations (400, and 800 mg/L), were used. The larvae (initial notochord length = 3.83 mm) were fed taurine-enriched rotifers in triplicate, from 3 days after hatching for 12 days. The average taurine contents of the rotifers were 0.31, 5.34, and 8.55 mg/g dry matter, respectively. The rotifers from all treatments had similar fatty acid composition. The growth and survival rates of the larvae fed rotifers enriched with 800 mg/L taurine supplementation were significantly higher than those of larvae fed rotifers without taurine enrichment (*p* = 0.005 and 0.002, respectively). The whole-body taurine content in the fish increased significantly with the increase in taurine level in the rotifers: 1.02, 3.48, and 4.11 mg/g in larvae fed control rotifers, and rotifers enriched with 400, and 800 mg/L taurine supplementation, respectively. The results of this study indicate that small yellow croaker larvae benefit from taurine concentrations above those typically reported in non-taurine-enriched rotifers.

## 1. Introduction

The small yellow croaker *Larimichthys polyactis*, a member of the family Sciaenidae, is a benthopelagic fish species widely found in temperate waters of the Northwest Pacific, including the Bohai Sea, Yellow Sea, and East China Sea [1]. In Korea, they spawn in mudflats from mid-March to June and are mostly caught in the Yellow Sea [2]. The small yellow croaker is a popular edible fish owing to its high nutritional value and soft texture [1,3]. It has also drawn substantial attention in Korea and China for its high commercial value [4], which has been increasing annually since the 1970s; the global catch volume reached 310,125 t in 2020 [5]. Moreover, the annual catch volume in the East China Sea recently exceeded 12,000 tons [6]. Consequently, small yellow croaker stocks have been significantly reduced. To overcome this, it is necessary to maintain the current stock and increase their productivity. Various contributions to develop artificial seed production and aquaculture technologies for small yellow croaker are being made, including studies on the female reproductive cycle [7], artificial propagation [8], and the dietary protein and lipid levels of juveniles [9]. However, studies on the improvement of growth and survival during the early larval stage of artificial seed production are lacking.

Growth and development occur rapidly during the early life stages of the fish; this is mainly temperature-dependent. Morphological and functional changes occur in many organs during metamorphosis, increasing the nutritional requirements per unit body mass at this time to more than those of adult fish [10]. Rotifers are used as the initial feed for larvae during the artificial seed production of many marine fish, including the small yellow croaker; they offer the advantages of being easy to mass produce and suitable in size for larvae to consume. Accordingly, rotifers are used as an initial feed during seed production. However, rotifers are deficient in certain nutrients, which can cause early mortality and poor growth among larvae and juveniles [11,12]. For example, commercial rotifers, generally supplied to hatchlings during seed production, contain lower concentrations of taurine and several micronutrients, such as iodine and selenium, than copepods, an initial food source found in nature [13,14].

Studies on the nutritional enrichment of rotifers have been conducted to address these nutritional shortcomings, and the importance of taurine enrichment has been emphasized in recent years. According to Takahashi et al. (2005) [15] and Takeuchi (2009) [16], the taurine content in hatched larvae that consumed rotifers decreased rapidly in accordance with the growth and development of the fish. In other words, fish have a high taurine requirement during the larval stage, suggesting the need for higher taurine contents in supplied rotifers [17,18]. However, the taurine content in rotifers has been reported to be 10–400 times lower than that in *Artemia* and copepods [13,14,19,20].

The physiological function of taurine (2-aminoethanesulfonic acid) has been studied in various vertebrates, including fish. Many animals synthesize taurine from methionine, cysteine, and 4’-phosphopantetheine through a series of enzymatic conversions [18,21]. Two key enzymes required for taurine synthesis process, namely, cysteamine dioxygenase and cysteine sulfinate decarboxylase, are less active in fish than in mammals, and taurine biosynthesis ability varies greatly depending on the fish species [21,22]. In other words, many marine fish larvae and juveniles cannot synthesize sufficient amounts of taurine, which is problematic because it is an essential nutrient in various life stages of marine fish [23,24]. Although taurine is not involved in protein synthesis due to its lack of a carboxyl group [25], it is still a vital nutrient with various physiological roles throughout the body [26]. Taurine’s primary functions have been identified to be the improvement of lipid digestion, antioxidant absorption, and osmoregulation, as well as the promotion of retina, nerve, and muscle tissue development [27,28]. In particular, taurine has been reported to play another role as an antioxidant. Given that decreased cholesterol synthesis, decreased disease resistance, impaired metamorphosis, and oxidative stress were observed when fish consume food containing an insufficient amount of taurine, a deficiency of taurine can greatly affect the growth performance and survival of fish [29,30]. Dietary taurine supplementation alters muscle fatty acid composition and effectively manages muscle oxidative stress [31].

Previous studies have demonstrated the benefits of feeding taurine-enriched rotifers to larvae of Japanese flounder *Paralichthys olivaceus* [32], Pacific cod *Gadus microcephalus* [33], amberjack *Seriola dumerili* [34], northern rock sole *Lepidopsetta polyxystra* [35], red sea bream *Pagrus major* [10,36], California yellowtail *Seriola lalandi* [37], Pacific bluefin tuna *Thunnus orientalis*, yellowfin tuna *Thunnus albacares* [38], and walleye pollock *Gadus chalcogrammus* [39]. However, to date, no study has focused on small yellow croakers. As a part of the effort to establish a seed production technology for small yellow croakers, this study evaluated the effects of varying concentrations of taurine-enriched rotifers on the growth and survival rate of small yellow croaker larvae.

## 2. Materials and Methods

Rearing of the experimental fish and sample collection were carried out at the Jeju Fisheries Research Institute (Jeju, Korea) of the National Institute of Fisheries Science (NIFS) of Korea in accordance with the Guidelines for Experimental Animals of NFIS Institutional Animal Care Use Committee (2021-NIFS-IACUC-21).

### 2.1. Fertilized Eggs and Rearing Conditions

Naturally spawned, fertilized eggs were obtained from a female small yellow croaker (3 years old, weight ≈ 200 g) and reared and managed at the Jeju Fisheries Research Institute of the NIFS. Next, fertilized eggs were distributed (10,000 eggs per tank) to nine 150 L circular polythene indoor tanks (hatching rate of 90.2 ± 5.1%, *n* = 3). Three experimental groups were prepared: a non-taurine-enriched rotifer feeding group (control) and two taurine-enriched rotifer feeding groups (T-400 and T-800). The concentration of taurine was determined based on previous studies [10,34,38]. The experiments were repeated in triplicate for each group (three tanks per experimental group). Control (non-enriched rotifers) and nutrient-enriched (400, and 800 mg/L taurine) rotifers were supplied to open-mouth hatched larvae (3 days after hatching (DAH)) as the primary feed up to 15 DAH. The water was exchanged at 100% of the tank per day with seawater filtered using a sand filter and ultrafilter membrane. In addition, 240 mL of commercial *Chlorella* (Fresh Chlorella V12, 14 milliard cell/mL; Chlorella Industry Co., Ltd., Tokyo, Japan) was added to the water tank thrice daily to prevent rotifer starvation and to maintain green water conditions. The average rearing water temperature, dissolved oxygen level and pH during the experimental period were 21.2 ± 0.5 °C, 7.3 ± 0.2 mg/L and 8.1 ± 0.1, respectively. The photoperiod was controlled from 09:00 to 21:00 h using a fluorescent lamp (12 L: 12 D).

### 2.2. Rotifer Cultivation and Taurine Enrichment

Five billion L-type rotifer (*Brachionus plicatilis*) individuals were used as the primary feed in the experiment. They were placed in a 1000 L cylindrical polypropylene tank and cultivated using an automatic feeder (AX1-51; Cheonsei Industry, Co., Ltd., Ansan, Korea) that supplied commercial *Chlorella* (Fresh Chlorella V12; Chlorella Industry Co., Ltd., Tokyo, Japan)—diluted 5-fold with seawater—at a rate of 5 mL/min. The cultivated rotifers were dispensed to three 20 L, acrylic, round water tanks (500,000 individuals/L in each), to which 20 mL of commercial docosahexaenoic acid (DHA) rotifer enricher (Marine Glos EX; Marine Tech, Aichi, Japan) and 400 or 800 mg/L of synthetic taurine (food and feed grade 99.0%; Qianjiang Yongan Pharmaceutical Co., Ltd., Yongan, China) were added for rotifer enrichment. During this process, the water temperature was set to 25 °C. Rotifer enrichment was carried out twice per day as follows: at 05:00 and 22:00 initially, followed by 10:00 and 17:00, and repeated thereafter (12 h between the last feed of the day and first feed of the proceeding day). To ensure adequate supply of rotifers to feed the small yellow croaker larvae, only the number of rotifers needed was collected from each nutrient-enriched tank, and they were washed in seawater at 25 °C for 5 min before feeding. The number of rotifers in the rearing tank was maintained at 1 individual/mL by checking the density before feeding.

### 2.3. Sampling and Measurement of Rotifers and Hatched Larvae

Rotifers collected before enrichment (0 h) and 12 h after enrichment were washed with seawater at 25 °C for 5 min and tap water for 3 min. After drying, all rotifer samples were stored at −80 °C until analysis. Small yellow croaker larvae were randomly sampled every 5 days from 0 to 15 DAH for use in growth performance and chemical analyses. The notochord length and body weight were measured by randomly sampling 20 individuals per tank at 0, 5, and 10 DAH and 50 individuals per tank at 15 DAH. The notochord length was measured using a stereoscopic microscope (SMZ745T; Nikon, Tokyo, Japan) and imaging software (Optiview; Korea Lab Tech, Seongnam, Korea), while the body weight was measured to the nearest 0.1 μg using a microscale (Quintix124-1SKR; Sartorius, Goettingen, Germany). All individuals in each tank were sampled at 15 DAH—the end of the experiment—for analysis of fatty acids and free amino acids, and the survival rate was calculated by counting the number of viable individuals. The weight gain (WG), specific growth rate (SGR), and survival rate were calculated as follows:WG (%) = [final wet weight (g) − initial wet weight (g)]/initial wet weight (g) × 100(1)
SGR (%/day) = [ln(final weight) − ln(initial weight)]/days × 100(2)
Survival rate (%) = 100 × [total number of larvae at 15 DAH + sampled larvae number]/[contained eggs × (egg hatching percent/100)](3)

### 2.4. Chemical Analysis

The collected rotifer and small yellow croaker larvae samples were freeze-dried in a freeze dryer (LP-20; Ilshin BioBase, Yangju, Korea) for at least 72 h. Subsequently, the samples (0.6 g, dry matter) were finely ground; free amino acids and fatty acids were quantified. Free amino acids were extracted three times repetitively by mixing one volume of the sample (0.1 g, dry matter) with 10 volumes of distilled water, 0.1 M of perchloric acid, and 0.1% meta-phosphoric acid, and heating the mixture in a boiling water bath. Through a centrifugation at 4000× *g* for 15 min (Supra 22, Hanil Scientific Inc., Gimpo, Korea), only supernatant was saved. To remove fat from this supernatant, we added an additional step that uses ether when distilled water and 0.1% meta-phosphoric acid were extracted. The resultant supernatant was concentrated using a rotary vacuum evaporator (Rotavapor R-124, Büchi, Switzerland), dissolved in 0.02 N of HCl, and filtered through a 0.45 μm membrane filter. Free amino acids were analyzed using a high-performance liquid chromatography system (Dionex Uitimate 3000 system; Thermo Scientific, Vienna, Austria) and a column (Inno C18 column; 4.6 × 150 mm, 5 μm; Younjinbiochrom, Co., Ltd., Seongnam, Korea). The analysis was performed under the following conditions: injection volume of 0.5 μm, column temperature of 40 °C, sample temperature of 20 °C, and flow rate of 1.5 mL/min. For buffer A (mobile phase A), 40 mM of sodium phosphate (pH 7) was used, while for buffer B (mobile phase B), a mixture of ultrapure water, acetonitrile, and methanol (10:45:45, *v*/*v*%) was used. Fatty acids were analyzed using a gas chromatography apparatus (Agilent 7890A; Santa Clara, CA, USA) and a column (DB-23; 120 × 0.25 mm; 0.25 μm; Agilent, Santa Clara, CA, USA). The amount of sample used for the analysis was 0.1 g, dry matter. The analysis was performed under the following conditions: injector temperature of 250 °C, flame ionization detector temperature of 280 °C, H_2_ concentration of 35 mL/min, air concentration of 350 mL/min, and He concentration of 10 mL/min.

### 2.5. Statistical Analysis

The growth performance and chemical analysis results were analyzed using IBM SPSS 19 (SPSS Inc., Chicago, IL, USA). The data were analyzed for normality and homogeneity of variances using Shapiro–Wilk’s and Levene’s tests, respectively, and showed no violation of the assumptions (*p* > 0.05). Statistical data analysis was performed using analysis of variance with the significance level set to 95% (*p* < 0.05). When a significant treatment effect was detected, the significant difference between the mean values was evaluated using Tukey’s honest significant difference test.

## 3. Results

### 3.1. Chemical Composition in Rotifer

Table 1 shows the free amino acid content of rotifers according to the concentration of taurine enrichment. Initial rotifer samples (pre-enrichment) showed no difference in taurine content compared to those in the control group (no enrichment) (0.31 mg/g), but showed a significant difference compared to those in the T-400 and T-800 groups (*p* = 0.000). The taurine content in rotifers varied according to the amount of taurine enrichment. The taurine content in the T-400 group was 17 times higher than that in the control group (5.34 mg/g), while that in the T-800 group was 27 times higher than that in the control group (8.55 mg/g). No taurine was detected in the *Chlorella* used as feed for rotifers (Table 1).

Table 2 shows the fatty acid composition of each experimental group. No DHA was detected in the fatty acids of the *Chlorella*. The DHA content in the fatty acids of the rotifers in the initial group without DHA enrichment was 0.28%, while that in the control, T-400, and T-800 groups was 12.12%, 11.27%, and 10.87%, respectively, which confirmed the enrichment effect of DHA (Table 2).

### 3.2. Growth Performance

Table 3 and Figure 1 show the growth performance of small yellow croaker larvae after the rearing experiment. In the 15 DAH larva samples, the T-800 group showed the longest notochords, followed by the T-400 and control groups, with a significant difference between the T-800 and control groups (*p* = 0.040). In addition, the weight gain (WG) and specific growth rate (SGR) were significantly higher in the T-400 and T-800 groups than in the control group (*p* = 0.006 and 0.007, respectively). Among the taurine-enriched groups, the T-800 group showed greater growth than the T-400 group, but the difference was not significant. The T-800 group also showed a significantly higher survival rate than the T-400 and control groups (*p* = 0.002).

### 3.3. Chemical Composition in Larvae

Table 4 shows the free amino acid contents in small yellow croaker larvae before and at the end of the experiment. The taurine content in the larvae showed a tendency to decrease as the rearing experiment progressed in all three groups (1.02 to 4.11 mg/g) compared to that at the start of the experiment (7.28 mg/g). The taurine content in the experimental groups showed a significant difference according to the level of taurine enrichment in the supplied rotifers, with the T-800 group showing the highest level of taurine accumulation, followed by the T-400 and control groups (4.11, 3.48, and 1.02 mg/g, respectively) (*p* = 0.000). Table 5 shows the fatty acid content in the small yellow croaker larvae before and at the end of the experiment. The DHA content was high in all experimental groups at the end of the experiment (22.83% to 23.41%) compared to that at the start of the experiment (16.81%). However, there was no significant difference in DHA and long-chain polyunsaturated fatty acid (LC-PUFA) content among the experimental groups.

## 4. Discussion

The direct enhancement of rotifers with taurine was effective in increasing the accumulated taurine concentration in rotifers, resulting in improved growth and survival rates in small yellow croaker larvae. Our proposed rotifer taurine enrichment period was 12 h, wherein enrichment at a concentration of 800 mg/L (T-800 group) showed the highest taurine accumulation (8.55 mg/g). Previous studies using the same taurine concentration (800 mg/L) reported that taurine concentrations in rotifers vary according to the enrichment period. Kim et al. (2016) [10] used 800 mg/L of taurine to enrich S-type rotifers *Brachionus rotundiformis* for 16 and 23 h. The taurine concentration in rotifers was 9.11 mg/g after 16 h of enrichment, which increased to 15.57 mg/g after 23 h of enrichment (initial taurine concentration before enrichment: 2.31 mg/g). A study by Chen et al. (2005) [32] used 800 mg/L of taurine to enrich L-type rotifers for 6 and 17 h. The taurine concentration was 2.41 mg/g after enrichment for 6 h and 4.52 mg/g after enrichment for 17 h, showing that the taurine content accumulated in rotifers increased with an increase in enrichment period (initial taurine concentration before enrichment: 1.09 mg/g). Another study by Chen et al. (2004) [36] compared taurine accumulation after the enrichment of S- and L-type rotifers for 6 and 17 h. The taurine accumulation in both the S- and L-type rotifers was higher after 17 h than that after 6 h of enrichment, but there was no significant difference between the taurine accumulation in S- and L-type rotifers. Both the aforementioned studies revealed the taurine concentration in rotifers to increase with long-term enrichment, while also showing that taurine concentrations may vary depending on the pre-enrichment concentration and enrichment conditions. Although this study only showed the results of taurine concentration after a single enrichment period (12 h), the results were similar to the taurine concentration in copepods in nature (8.32 mg/g), as reported by Yamamoto et al. (2008) [40]. Despite the shorter cultivation time than those used in other studies, the results indirectly confirmed that the enrichment period used was effective based on the high taurine accumulation.

Most previous studies on taurine enrichment in rotifers showed increased taurine accumulation with increased taurine concentrations used for enrichment. A study by Matsunari et al. (2013) [34] also showed that taurine accumulation in rotifers increased to 3.2, 4.2, 5.2, and 7.4 mg/g after enrichment for 9 h using 0, 200, 400, and 800 mg/L of taurine, respectively. Similarly, this study showed taurine accumulation in rotifers increasing by up to 5.34 and 8.55 mg/g after enrichment using 400 and 800 mg/L of taurine, representing a 17- and 27-fold increase, respectively, compared to the taurine accumulation in non-enriched rotifers (0.31 mg/g). These results suggest the need for enrichment with a higher taurine concentration during seed production for species with a high taurine requirement or during the early life development stages. Another factor to consider during taurine enrichment in rotifers is the taurine accumulation efficiency of rotifers according to the added concentration. Results from the current and previous studies revealed that the taurine accumulation in rotifers to increases with an increase in taurine enrichment concentration but that the taurine accumulation efficiency per unit time decreases. Therefore, it is necessary to determine the nutritional taurine requirement of the target species through investigation of growth performance with respect to taurine accumulation for taurine enrichment in rotifers. Because the objective of this study was to investigate the effects of taurine on the growth performance and survival rate of small yellow croaker larvae, the taurine requirement during this stage was not investigated. However, additional future studies are planned to identify the nutritional requirement, the appropriate taurine enrichment concentration and optimal enrichment period according to such requirements.

During the early larval stage, the nutritional requirements to support rapid growth and development are high. Moreover, as biosynthetic ability is incomplete during this stage, an imbalance between the external exogenous supply of taurine and the nutritional requirements of the body can lead to significant problems, such as poor growth [10,34,38]. Rotifers actually contain a low level of taurine compared to other food sources (i.e., *Artemia* sp. and copepods), and studies have suggested that a single long-term supply of rotifers during the seed production period could further accelerate taurine deficiency in hatched larvae [16,19]. Accordingly, various studies on the improvement of growth and development during the early larval stage through taurine supply have been conducted on various fish species to address this problem [35,41,42]. A study by Katagiri et al. (2017) [38] reported that when rotifers were supplied to Pacific bluefin tuna and yellowfin tuna for 16 days starting from 2 DAH, a significant improvement in growth performance was observed in the taurine-enriched group. Similarly, Matsunari et al. (2013) [34] reported improved growth and survival rate after supplying taurine-enriched rotifers to amberjack larvae. Rotman et al. (2017) [37] found that feeding California yellowtail with taurine-enriched rotifers resulted in a significant improvement in growth performance. Contrary to these previous study results, there was no significant difference in growth among white seabass *Atractoscion nobilis* fed taurine-enriched rotifers, which suggested that the nutrient enrichment effects may vary depending on species [37]. This study found a significant improvement in the growth performance of small yellow croaker larvae after taurine enrichment. In particular, the T-800 group showed a significantly higher growth performance and better survival rate than the control group. In general, physiological responses to nutritional deficiencies tend to occur readily during the early life stages of fish species that depend on exogenous nutrients and have low nutrient accumulation rates and high metabolic rates [43,44]. Further, larvae have a high metabolism rate and thus they are susceptible to oxidative stress, considering high amounts of LC-PUFA [45,46]. In this study, the group supplemented with taurine (T-400 and T-800) showed significantly higher survival rates than its control group, suggesting that the exogenous taurine may have increased the antioxidant capacity and immune functions of the small yellow croaker larvae. Similar studies reported that the supplementals of taurine led to the higher immunity and antioxidant capacity of black carp *Mylopharyngodon piceus* [47], rice field eel *Monopterus allbus* [48], California yellowtail *Seriola dorsalis* [49], grass carp *Ctenopharyngodon idella* [50], and European seabass *Dicentrarchus labrax* [51]. It is supposed that small yellow croakers need supplementation through exogenous nutrients, owing to their low ability to biosynthesize taurine. The findings of this study, which revealed an improvement in growth after supplementation of taurine-enriched rotifers, suggest that taurine is an effective nutrient during the larval stage of small yellow croakers.

It is known that one of the major factors that can cause larval mortality during seed production is malnutrition, which occurs after yolk absorption. Larvae and adult fish take different times for nutrient digestion, absorption, and organ development; these are temperature-dependent. Although this is not a general rule for all fish larvae, most larvae ingest food through pinocytotic ingestion via the rectal epithelial cells without gastric activity until 10 DAH. Red sea bream has less endogenous nutrients than other marine fish species; however, it is capable of transitioning smoothly from endogenous nutrient intake from the yolk to exogenous nutrients by feeding, owing to which it has a higher early feeding capability [52]. Moreover, free amino acids in the yolk are used as essential nutrients during the egg development stage of Paralichthyidae and Pleuronectidae; 70% are consumed as energy, while the remaining 30% are converted to body composition proteins, i.e., they are used as an energy source comparable to lipids even after hatching [53,54]. This suggests that addition of taurine during the larval stage (rotifer intake stage) is very effective for promoting larval growth and development. There were no significant differences in the eicosapentaenoic acid (EPA), DHA, and n-3 LC-PUFA contents in small yellow croaker larvae; similar values were noted. However, factors other than fatty acids, i.e., utilization efficiency of amino acids, were considered to be extremely high in all three experimental groups.

## 5. Conclusions

The findings in this study show that the natural taurine content in rotifers is insufficient for the rapid growth of the small yellow croaker larvae, and dietary supplementation with taurine-enriched rotifers effectively improves the growth and survival of larvae of this species. Such findings could be used to determine the physiological functions of taurine in the fish larval stage and to improve the mass production of small yellow croakers. Studies on long-term effects and the effects of higher levels of taurine enrichment (>800 mg/L) during the juvenile stage could elucidate the role of taurine enrichment in the successful aquaculture of this species.

## Figures and Tables

**Figure 1 animals-12-01403-f001:**
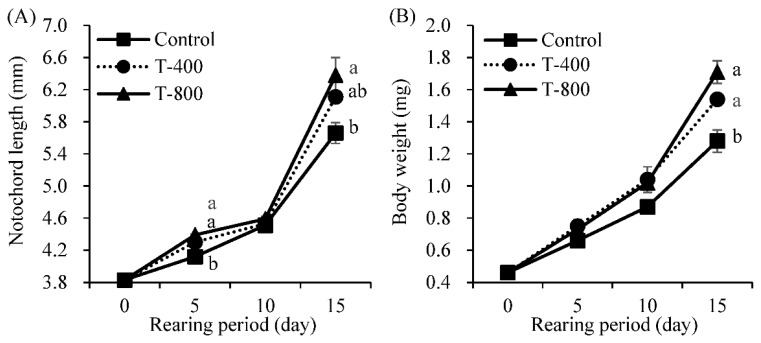
Notochord length ((**A**), *n* = 3 tanks) and body weight ((**B**), *n* = 3 tanks) of small yellow croaker fed different taurine-enriched rotifers. Vertical bar indicates the standard error of mean values of triplicate tanks. Different superscripts within each row are significantly different (*p* < 0.05) by Tukey’s test. Three different taurine (T) enrichment levels, including 400 mg/g (T-400) and 800 mg/g (T-800).

**Table 1 animals-12-01403-t001:** Free amino acid content of rotifers (mg/g, dry matter).

			Enriched Rotifer ^1^		
	*Chlorella*	Initial Rotifer	Control	T-400	T-800
Non-essential amino acids ^2^					
Taurine	N.D.	0.30 ± 0.00	0.31 ± 0.02 ^c^	5.34 ± 0.29 ^b^	8.55 ± 0.06 ^a^
Alanine	4.75 ± 0.14	0.86 ± 0.01	3.93 ± 1.10	6.19 ± 0.73	6.15 ± 0.66
Aspartic acid	0.16 ± 0.02	0.41 ± 0.01	1.50 ± 0.31	1.87 ± 0.48	1.24 ± 0.16
Glutamic acid	2.97 ± 0.08	1.67 ± 0.00	5.99 ± 1.53	7.22 ± 0.48	6.67 ± 0.23
Glycine	0.82 ± 0.04	0.47 ± 0.01	1.82 ± 0.42	2.71 ± 0.36	2.59 ± 0.28
Proline	1.77 ± 0.12	0.37 ± 0.00	1.84 ± 0.63	3.13 ± 0.45	3.21 ± 0.42
Serine	0.11 ± 0.02	0.48 ± 0.01	1.53 ± 0.24	2.17 ± 0.18	1.67 ± 0.01
Tyrosine	0.18 ± 0.03	0.84 ± 0.02	1.80 ± 0.41	1.65 ± 0.31	1.40 ± 0.08
Total	10.77 ± 0.44	5.41 ± 0.04	18.71 ± 4.57	30.28 ± 1.72	31.46 ± 1.30
Essential amino acids ^2^					
Arginine	0.34 ± 0.04	1.58 ± 0.03	3.12 ± 0.52	2.92 ± 0.58	1.91 ± 0.29
Histidine	0.14 ± 0.02	0.47 ± 0.02	1.06 ± 0.24	1.28 ± 0.14	1.13 ± 0.06
Isoleucine	0.23 ± 0.02	0.34 ± 0.01	2.87 ± 0.97	4.84 ± 0.80	4.73 ± 0.63
Leucine	0.18 ± 0.06	0.66 ± 0.01	4.97 ± 1.63	8.12 ± 1.20	7.89 ± 0.99
Lysine	0.48 ± 0.04	0.84 ± 0.03	2.54 ± 0.57	2.42 ± 0.51	1.62 ± 0.19
Methionine	0.04 ± 0.02	0.18 ± 0.01	1.09 ± 0.38	1.75 ± 0.23	1.66 ± 0.21
Phenylalanine	0.17 ± 0.04	0.53 ± 0.02	3.07 ± 0.99	4.84 ± 0.64	4.78 ± 0.61
Threonine	0.38 ± 0.02	0.39 ± 0.01	2.32 ± 0.72	3.81 ± 0.51	3.69 ± 0.43
Valine	0.33 ± 0.03	0.34 ± 0.01	2.54 ± 0.86	4.28 ± 0.69	4.18 ± 0.53
Total	2.28 ± 0.29	5.34 ± 0.12	23.58 ± 6.78	34.26 ± 3.55	31.58 ± 3.00
Total (Non-essential + essential)	13.04 ± 0.72	10.74 ± 0.16	42.30 ± 11.35	64.54 ± 4.98	63.05 ± 4.30

^1^ Three different taurine (T) enrichment levels, including 400 mg/g (T-400) and 800 mg/g (T-800). N.D., not detected. ^2^ Values are means ± standard deviation of triplicate (*n* = 3). ^a,b,c^ Values in the taurine row among enriched rotifer groups with different superscript letters are significantly different (Tukey’s test, *p* < 0.05).

**Table 2 animals-12-01403-t002:** Fatty acid composition (% of total fatty acids) of rotifers.

			Enriched Rotifer ^1^		
	*Chlorella*	Initial Rotifer	Control	T-400	T-800
Fatty acid (area, %) ^2^					
ΣSFA ^3^	26.16 ± 0.07	34.60 ± 0.39	43.50 ± 0.32	43.82 ± 0.32	45.00 ± 0.92
ΣMUFA ^4^	4.92 ± 0.03	6.12 ± 0.06	5.79 ± 0.09	6.19 ± 0.09	6.19 ± 0.08
18:2n-6	63.64 ± 0.07	47.85 ± 0.29	27.40 ± 0.32	27.69 ± 0.16	27.28 ± 0.52
20:2n-6	N.D.	3.42 ± 0.05	1.55 ± 0.01	1.60 ± 0.01	1.54 ± 0.03
20:3n-6	N.D.	3.12 ± 0.06	1.60 ± 0.04	1.57 ± 0.05	1.54 ± 0.05
20:4n-6	N.D.	1.75 ± 0.05	2.13 ± 0.03	1.99 ± 0.04	1.91 ± 0.06
Σn-6PUFA ^5^	63.64 ± 0.07	52.73 ± 0.39	31.13 ± 0.37	31.25 ± 0.24	30.73 ± 0.62
18:3n-3	4.91 ± 0.02	2.55 ± 0.03	3.75 ± 0.02	3.84 ± 0.04	3.77 ± 0.10
20:5n-3 (EPA)	N.D.	0.29 ± 0.02	2.16 ± 0.01	2.03 ± 0.05	1.90 ± 0.05
22:6n-3 (DHA)	N.D.	0.28 ± 0.07	12.12 ± 0.21	11.27 ± 0.41	10.87 ± 0.20
Σn-3PUFA	4.91 ± 0.02	3.12 ± 0.06	18.04 ± 0.20	17.14 ± 0.48	16.54 ± 0.34
Σn-3LC-PUFA ^6^	N.D.	0.57 ± 0.07	14.28 ± 0.22	13.30 ± 0.45	12.77 ± 0.25

^1^ Three different taurine (T) enrichment levels, including 400 mg/g (T-400) and 800 mg/g (T-800). N.D., not detected. ^2^ Values are means ± standard deviation of triplicate (*n* = 3). ^3^ SFA, saturated fatty acids include 14:0, 16:0, 17:0, 18:0, 20:0, 22:0 23:0, 24:0; ^4^ MUFA, monounsaturated fatty acids include 16:1n-7, 18:1n-9, 22:1n-9, 24:1n-9; ^5^ PUFA, polyunsaturated fatty acids; ^6^ LC-PUFA, long-chain polyunsaturated fatty acids.

**Table 3 animals-12-01403-t003:** Growth performance of yellow croaker larvae fed with the various taurine-enriched rotifers for 15 days ^1^.

Measurement	Enriched Taurine (mg/L)			*p*-Value
	Control	T-400	T-800	
Growth performance ^2^				
Final notochord length (mm)	5.66 ± 0.13 ^b^	6.11 ± 0.04 ^a,b^	6.38 ± 0.22 ^a^	0.040
Final body weight (mg)	1.28 ± 0.07 ^b^	1.54 ± 0.02 ^a^	1.71 ± 0.07 ^a^	0.005
Weight gain (%)	178.42 ± 15.78 ^b^	226.09 ± 5.12 ^a^	272.46 ± 14.44 ^a^	0.006
Specific growth rate (%/day)	6.80 ± 0.39 ^b^	8.06 ± 0.10 ^a^	8.76 ± 0.26 ^a^	0.007
Survival (%)	9.26 ± 0.24 ^b^	10.36 ± 0.34 ^b^	12.57 ± 0.16 ^a^	0.002

^1^ Mean ± standard error (*n* = 3 tanks) with different superscripts within each row are significantly different (Tukey’s test, *p* < 0.05). ^2^ Initial notochord length = 3.83 mm (*n* = 30), body weight = 0.46 mg (*n* = 30).

**Table 4 animals-12-01403-t004:** Free amino acid content of small yellow croaker larvae fed with the various taurine-enriched rotifers for 15 days (mg/g, dry-matter).

	Initial	Final ^1^			*p*-Value
		Control	T-400	T-800	
Non-essential amino acids ^2^					
Taurine	7.28 ± 0.18	1.02 ± 0.20 ^c^	3.48 ± 0.26 ^b^	4.11 ± 0.15 ^a^	0.000
Alanine	3.64 ± 0.06	0.33 ± 0.08	0.19 ± 0.03	0.20 ± 0.03	0.053
Aspartic acid	0.17 ± 0.02	0.25 ± 0.04 ^a^	0.18 ± 0.02 ^a,b^	0.17 ± 0.02 ^b^	0.041
Glutamic acid	1.15 ± 0.01	1.05 ± 0.09 ^a^	0.72 ± 0.05 ^b^	0.77 ± 0.05 ^b^	0.002
Glycine	1.25 ± 0.01	0.21 ± 0.02 ^a^	0.15 ± 0.02 ^b^	0.14 ± 0.02 ^b^	0.014
Proline	0.38 ± 0.02	0.09 ± 0.01 ^a^	0.07 ± 0.00 ^b^	0.07 ± 0.01 ^b^	0.008
Serine	1.52 ± 0.02	0.34 ± 0.02 ^a^	0.24 ± 0.04 ^b^	0.22 ± 0.03 ^b^	0.008
Tyrosine	1.35 ± 0.01	0.18 ± 0.03	0.13 ± 0.01	0.13 ± 0.04	0.203
Total	16.74 ± 0.17	3.46 ± 0.23 ^c^	5.17 ± 0.13 ^b^	5.82 ± 0.24 ^a^	0.000
Essential amino acids ^2^					
Arginine	0.61 ± 0.03	0.12 ± 0.01	0.11 ± 0.00	0.11 ± 0.02	0.680
Histidine	1.20 ± 0.02	0.13 ± 0.02	0.10 ± 0.01	0.10 ± 0.02	0.198
Isoleucine	1.40 ± 0.05	0.08 ± 0.01	0.06 ± 0.01	0.06 ± 0.01	0.173
Leucine	2.19 ± 0.03	0.16 ± 0.02	0.14 ± 0.02	0.14 ± 0.03	0.388
Lysine	1.15 ± 0.03	0.11 ± 0.01	0.10 ± 0.01	0.11 ± 0.02	0.630
Methionine	0.56 ± 0.03	0.09 ± 0.01	0.09 ± 0.01	0.09 ± 0.01	0.927
Phenylalanine	1.95 ± 0.02	0.12 ± 0.02	0.11 ± 0.01	0.12 ± 0.03	0.927
Threonine	0.92 ± 0.02	0.16 ± 0.01	0.14 ± 0.01	0.15 ± 0.02	0.296
Valine	2.60 ± 0.02	0.13 ± 0.01	0.10 ± 0.01	0.10 ± 0.02	0.252
Total	12.59 ± 0.14	1.10 ± 0.10	0.95 ± 0.08	0.99 ± 0.19	0.441
Total (Non-essential + essential)	29.33 ± 0.31	4.56 ± 0.13 ^c^	6.11 ± 0.10 ^b^	6.81 ± 0.36 ^a^	0.000

^1^ Three different taurine (T) enrichment levels, including 400 mg/g (T-400) and 800 mg/g (T-800). ^2^ Values are means ± standard deviation of triplicate (*n* = 3). Values in the same row with different superscript letters are significantly different (Tukey’s test, *p* < 0.05).

**Table 5 animals-12-01403-t005:** Fatty acid composition (% of total fatty acids) of small yellow croaker larvae fed with various taurine-enriched rotifers for 15 days.

	Initial	Final ^1^			*p*-Value
		Control	T-400	T-800	
Fatty acid (area, %) ^2^					
ΣSFA ^3^	25.35 ± 0.22	41.53 ± 0.37	42.04 ± 0.64	41.80 ± 1.16	0.746
ΣMUFA ^4^	40.92 ± 0.13	8.76 ± 0.30	9.16 ± 0.45	8.55 ± 0.03	0.844
18:2n-6	7.19 ± 0.02	16.20 ± 2.46	15.25 ± 1.17	16.28 ± 1.16	0.723
20:2n-6	0.24 ± 0.00	1.42 ± 0.13	1.34 ± 0.05	1.34 ± 0.05	0.430
20:3n-6	N.D.	1.47 ± 0.17	1.41 ± 0.11	1.52 ± 0.08	0.549
20:4n-6	1.32 ± 0.01	4.71 ± 0.15	4.63 ± 0.08	4.64 ± 0.16	0.761
Σn-6PUFA ^5^	8.52 ± 0.03	22.38 ± 2.49	21.29 ± 1.32	22.45 ± 1.08	0.678
18:3n-3	1.00 ± 0.01	1.06 ± 0.26	0.92 ± 0.14	1.07 ± 0.11	0.560
20:5n-3	7.16 ± 0.09	2.01 ± 0.11	1.84 ± 0.13	1.92 ± 0.04	0.189
22:6n-3	16.81 ± 0.24	22.83 ± 3.08	23.41 ± 0.94	22.86 ± 2.37	0.943
Σn-3PUFA	24.97 ± 0.31	25.90 ± 2.73	26.17 ± 0.79	25.85 ± 2.31	0.981
Σn-3LC-PUFA ^6^	23.97 ± 0.33	24.84 ± 2.98	25.25 ± 0.84	24.79 ± 2.41	0.963

^1^ Three different taurine (T) enrichment levels, including 400 mg/g (T-400) and 800 mg/g (T-800). N.D., not detected. ^2^ Values are means ± standard deviation of triplicate (*n* = 3). Values in the same row with different superscript letters are significantly different (Tukey’s test, *p* < 0.05). ^3^ SFA, saturated fatty acids include 14:0, 16:0, 17:0, 18:0, 20:0, 22:0 23:0, 24:0; ^4^ MUFA, monounsaturated fatty acids include 16:1n-7, 18:1n-9, 22:1n-9, 24:1n-9; ^5^ PUFA, polyunsaturated fatty acids; ^6^ LC-PUFA, long-chain polyunsaturated fatty acids.

## Data Availability

Not applicable.

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
