# Peer review of "A Preliminary Study on the Effects of Taurine-Enriched Rotifers on the Growth and Survival of the Small Yellow Croaker Larimichthys polyactis Larvae"

_animals, 2022, doi:10.3390/ani12111403_

Round 1

Reviewer 1 Report

The authors investigated the effect of Taurine-enriched rotifers on the growth and survival of small yellow croaker Larimichthys polyactis larvae. They designed three treatments to test their hypothesis. This manuscript (MS) was clearly written and easy to understand. This work can help the sustainability of this species farming as few amnio acid types of research have been done. However, some minor issues significantly compromised the quality of this MS.

However, I have touched on some more points that can contribute to the improvement of this MS.

Minor comments

Abstract

  • Line 9. the effect of taurine-enriched rotifers on larval growth and survival in the small yellow croaker Larimichthys polyactis was investigated.
  • Line 19, as the highest level had the best performance, testing higher levels should be suggested for future studies. Please add this point to the conclusion and discussion section.

Introduction:

  • Well-developed introduction and included a clear fellow and relevant points.
  • Line 58-79, please summarise this part.
  • Line 79-84, please clarify which level has been optimum for these fishes.

Material and methods

  • Well-organized section. Clear fellow and all required details were provided.
  • Please mention how many percentages of water were exchanged each day if you have monitored.
  • For each analysis, please clarify how many fish were taken.

Results

  • Well-written section; all necessary things have been covered.
  • Table 3, please write measurements in the complete form.
  • Here and elsewhere, report P uppercase and italic (P<0.05).
  • Line 239, throughout the MS, if there is no significant difference, no need to report P-value.
  • Table 2 and Table 4, please add SFA, MUFA, PUFA.
  •  

Discussion

  • Line 315 and elsewhere, please first mention the common name plus scientific name, and for the rest of the MS, just report the common name.
  • Some parts of the discussion are better updated with research in 2021 and 2022 as they refer to some old references. Please update the discussion with the latest studies as much as possible.
  • Although you wrote this section well, you can still improve it by answering these questions and annotated them in the discussion section. Why were these results observed? Discuss more possible reasons.
  • The conclusion needs to be revised and add more comprehensive concepts there.

Best regards

Author Response

Dear. Reviewer 1.

We have responded to the comments from the Reviewer 1 (see attached in a separate file) and we hope our responses are adequate for our paper to be accepted for publication.

Sincerely Yours,

Jin Woo Park, Ph.D.

Reviewer 2 Report

General comments

The authors evaluated the effects of taurine in a control group (0 taurine) and 2 enriched concentrations (400 and 800 mg/L) in the live food (rotifers) for yellow croaker larvae and investigated the effect of the amino acid in the survival and growth of the larvae.

The article contain good information regarding the importance of taurine for the studied species, anyway I have some concerns with the manuscript. The study provided one amino acid for the fish and evaluated the effect on survival, growth, the amino acid and fatty acids profile observed in the rotifers and in the fish larvae. Evaluating the content of the amino acids in the rotifers is a basic step in order to have the proper amount of the taurine in the live feed that will be provided to the fish. Evaluate the survival, growth and amino acids profile of fish is a good approach to understand some responses of the fish larvae to the taurine enriched diet.

But what is the meaning of evaluating fatty acids profile of the larvae? It is meaningless this evaluation in this study, since the authors did not mention any important role or relation between taurine and fatty acids in fish larvae development. The authors could have evaluated many other aspects that are more relevant in the fish larvae physiology when fed with taurine. In the introduction the authors mention the importance of taurine in the fish lipid digestion, antioxidant absorption and osmoregulation, retina nerve and muscle tissue development, so, they should evaluate physiological parameters related to the effect of taurine and correlate it with the concentrations provided to the fish. In addition, they just mention one time one line 43 about the PUFA n-3 importance for the fish larvae development, which is strange, since if they evaluated the fatty acid profile of the fish they should explain better the role of fatty acids (mainly PUFA n-3) during fish larvae development.

The introduction is not so well organized and it should be mentioning more information about fatty acid profile, since the authors presented this results. The methods are described but some few information missing that can be easily added. The results are no well-presented and it needs improvement. The discussion is poor and not well described.

Briefly, I am concerned about the volume of results and the connection between what is the objective of the work, the tools used to evaluate and the obtained results. Some connections are missing and I am afraid that the amount of results are not satisfactory for the publication of the work. Also, the authors evaluated a very short period during larval rearing, which is reasonable from the point of view that they want to evaluate rotifers enrichment, but if this period is short they should use more physiological tools that could bring more strong physiological or morphological responses to the study. The discussion is very poor and very long without going deep to the observed results. The authors speak a lot about the taurine deposition in the rotifers and this are methods and not results. If they investigated the effect of taurine in the larvae, the method for providing this taurine must be establish, otherwise, they should try the methods and show the methods and results about which method of inclusion was better, but it is not the case of this research.

They authors should be careful on references also, there are 75 citations in total in this work and many of them are cited without any relevant information. If not mistaken, 11 references are from 80s and 90s and 11 references are between year 2000 and 2005. This is more than 17 years ago and is a representative number of citations. Please reduce the number of citations, 75 is too much and they are not being properly used.  Citations around 50 are enough for a well cited and discussed manuscript.

Specific comments about the study

Line 8 – Missing the simple summary

Line 9 – The abstract is not mentioning any information about the fatty acid analysis. In addition, the P value should be informed the real value, not that p is less than 0.05. If there is difference the authors should inform how different was, and it is shown by the p value. It is clear that is there is a difference it is less than 0.05.

Line 10 and 11 – The group with 0 mg/L should not be call an enriched group. This group should be called `control group`.

Line 29 – Why the natural stocks are being depleted? What are the main threatens? Overfishing? Please mention more information here and cite the authors.

Line 36/37 – The growth of larvae is very quick in almost all species and this is mainly temperature dependent, so rephrase it please. This is common sense.

Line 37 to 39 – The morphology and functionality of all fish changes during larval rearing. It is a period where multiple changes happens. The interesting point is how this change occur and why? Which organs are more relevant during the initial development? Which author mention that the requirements of the per unit body mass at larval period is higher than in adults? Please cite it.

Line 36 to 57 – This paragraph is too long and the phrases are not well linked with a sequence. Please rephrase it. The authors started with growth and development of larvae, say about rotifers, nutritional requirements end with the taurine effect. I think it should be more interesting if provided more information about nutritional requirements and the physiological role of taurine during larval phase and the importance to provide it during the larvae exotrophic phase, such as the other paragraph (line 58 to 67) since it is basic the rotifers importance in the fish larvae first feeding.

Line 42-43 – n-3 fatty acids are essential so they are not only important, the fish do not synthetize it (in the most part of fish species) so it must be provide in the diet. Also the citation (9 is from 1991 and 10 is from 1989, this is really very old references, there are much research and information on the omega 3 essentiality and larvae nutrition, please provide more recent publications than these.

Line 85-89 – These are not results, evaluating the content of amino acids of the rotifers is a basic step to make sure that the live feed had the amount of taurine to feed the fish. This has to be done since it is a nutritional study.

Line 98-99 – How you estimate the hatching rate if the larvae hatched in the tank?

Line 107- Which concentration of Chorella was add to the tank? Please provide the concentration.

Line 120 – There is twice time the Docosahexaenoic acid in the phrase. Please correct it.

Line 128 – Please provide the density in rotifers per ml. We can calculate as 150.000 per 150 liters it will be 1 per ml, but please express the data per ml. This is the normal method. Also, this was the density used during all the experiment? This is very low concentration.

Line 149 – Please provide the amount of sample used for the extraction.

Line 165 – Please provide the amount of samples used for the extraction.

Line 190 – Table should be improved, the signals of + and – are different presented, sometimes with space and sometimes without space, please format it. Also, the lines of the table are not well formatted. Please format it also.

Line 200 – Please provide the fatty acid profile per group of fatty acids saturated fatty acid (SFA), monounsaturated fatty acids (MUFA) and polyunsaturated fatty acid (PUFA).

Line 222 – Figure 1 has to figures, please identify it as “a” and “b”.

Line 246 – Table should be formatted properly. Also, provide the fatty acids as groups, so, show first all saturated, monounsaturated and polyunsaturated together.

253 to 278 – In all this first part of the discussion the authors discuss about levels and time of inclusion. This experiment was not performed for using better enrichments time or amount, so it is not interesting to have this so long discussion about this topic. If the result observed was good with 12 hours of enrichment and this was close to good results the method worked, but it is meaningless to discuss to much about it. If the authors wants to do tests about time of enrichment and show this result it is good, but in this experiment it is not necessary to discuss this. If the authors evaluated the larvae survival and growth when fed with enriched rotifers, the rotifers enrichment protocol must be established, otherwise they should do enrichment trials first and then larval nutrition trials. Enrichment in the case of this experiment are methods and not results.

279-300 – please this discussion is also meaningless if the main idea of the manuscript was to understand the growth and survival of the fish yellow croaker fed with taurine. This part of discussion can be reduce and inserted together with the previous paragraph. So please put both paragraphs together in order to optimize and increase the quality of the manuscript since there are almost 50 lines with poor discussion of the data.

323-328 – This phrase are very confused information. Also, this is the first mention to fatty acids in the discussion. If the authors presented results with fatty acids they should discuss it. What the author means with this phrase “Further, larvae stage has high metabolism rate and thus they are susceptible to oxidative stress, considering high amount of long chain polyunsaturated fatty acids long chain polyunsaturated fatty acid (LC-PUFA)”, It is not clear what kind of connection there is with oxidative stress and LC-PUFA.

328-331 – The result don’t show any evidence of increasing anti oxidative stress or immune system improvement, so this is speculation only, without strong basis. Speculation is fine when you have evidence, but this is not the case. This is one the concerns with the few results obtained by the authors, they cannot be conclusive since the results are not supportive for big assumptions.

336-338 – This study does not give enough information about essentiality of taurine for the studied species. The essentiality means that the nutrient must be given in the diet and the results in spite of showing increase in growth and survival, the difference between groups the survival of the group without supplementation was very low and vary on average from 9 to 12%. The growth difference was higher from 1.28 to 1.71g, but and in terms of production, this is a good result, even with only 15 days of study, but not enough to say that taurine is essential to the species.  

340-342 – Fish larvae in general only have gastric digestion with the presence of gastric glands. Some fish can develop very fast the gastric glands but few species has the potential to develop gastric glands within 10 days, so this is not valid for many species. Please insert that this is not a rule for all fish larvae. Also, organs development are temperature dependent, and this vary among species.

343-349 – Fish larvae development needs proteins and lipids since all cells only growth with these two structural substrates available. In addition, some amino acids and some fatty acids are essential for marine fish. Remembering that essentiality is a term used to define that the nutrient is not synthetized by the animal, so it must be provided in the feed. In addition, the demand is high since fish is growing very fast, so fish must have appropriate yolk sac content of nutrients. In addition, from the first feeding and during the transition to exogenous feeding, proteins, lipids, amino acids, fatty acids are extremely important, so each nutrient has its own importance and must be provided in the feed.

350-355 – Why the authors would expect a significant difference between groups in fatty acids profile since they vary an amino acid in the live feed.

The authors evaluated the amino acid profile and did not explore only one amino acid during the results or in the discussion section. Some differences were observed in the amino acids profile and in the fatty acid profile but no result presentation or discussion is presented. This is disappointing and reduced the quality of the findings.

Author Response

Dear. Reviewer 2.

We have responded to the comments from the Reviewer 2 (see attached in a separate file) and we hope our responses are adequate for our paper to be accepted for publication.

Sincerely Yours,

Jin Woo Park, Ph.D.

Round 2

Reviewer 2 Report

Dear all, 

I think the article quality improved a lot and the authors made a great effort in order to enhance the quality of the proposed improvements in the manuscript. My only main concern is that I don´t see any good reason for evaluating the fatty acid profile of the larvae if they varied taurine in the diet, anyway it is a new study that observed good correlation between the high amount of taurine and survival and growth of the larvae. 

Author Response

The authors sincerely thank Reviewer#2 for your efforts is improving the quality of this manuscript.

The response to the comments is provided below.

We agree with the reviewer. As in the previous our responses, we believe that there was little or no effect of the fatty acid profile on the growth and survival of small yellow croaker larvae. In future studies, we will consider to conduct experiments to evaluate the effect of the correlation between taurine content and fatty acids profile (especially LC-PUFA) on survival and growth of small yellow croaker larvae as per your suggestion.